# Urokinase-Type Plasminogen Activator Receptor (uPAR) Cooperates with Mutated *KRAS* in Regulating Cellular Plasticity and Gemcitabine Response in Pancreatic Adenocarcinomas

**DOI:** 10.3390/cancers15051587

**Published:** 2023-03-03

**Authors:** Luogen Peng, Yuchan Li, Sha Yao, Jochen Gaedcke, Victor M. Baart, Cornelis F. M. Sier, Albrecht Neesse, Volker Ellenrieder, Hanibal Bohnenberger, Frieder Fuchs, Julia Kitz, Philipp Ströbel, Stefan Küffer

**Affiliations:** 1Institute of Pathology, University Medical Center Göttingen, University of Göttingen, 37075 Göttingen, Germany; 2Department of Oncology, Changsha Central Hospital, University of South China, Changsha 410004, China; 3Department of Pathology, The 3rd Xiangya Hospital, Central South University, Changsha 410013, China; 4Department of General, Visceral and Pediatric Surgery, University Medical Center Göttingen, 37075 Göttingen, Germany; 5Department of Surgery, Leiden University Medical Center, 2333 ZA Leiden, The Netherlands; 6Department of Gastroenterology and Gastrointestinal Oncology, University Medical Center Göttingen, 37075 Göttingen, Germany; 7Department of Medicine, Israelitisches Krankenhaus, 22297 Hamburg, Germany; 8Department of Microbiology and Hospital Hygiene, Bundeswehr Central Hospital Koblenz, 56070 Koblenz, Germany; 9Institute for Medical Microbiology, Immunology and Hygiene, University of Cologne, Medical Faculty and University Hospital of Cologne, 50931 Cologne, Germany

**Keywords:** pancreatic cancer, uPAR, KRAS, FAK, MEK, ERK, dormancy, gemcitabine

## Abstract

**Simple Summary:**

Pancreatic ductal adenocarcinomas (PDACs) with gene amplification or overexpression of urokinase-type plasminogen activator receptor (uPAR) have a particularly dismal prognosis. We show here that uPAR reinforces the MEK/ERK signaling pathway in tumor cells with a *KRAS* mutation with the suppression of FAK and CDC42 signaling. This synergy keeps tumor cells in a mesenchymal state that favors cell migration and proliferation, but also sensitizes them towards gemcitabine. These observations highlight a potential therapeutic dilemma that applies to KRAS and uPAR as emerging targets. Treatments targeting either KRAS or uPAR could induce cellular dormancy and render the tumor more resistant to chemotherapy (such as gemcitabine). The clinical benefit of adding autophagy inhibitors such as chloroquine in this situation remains to be shown.

**Abstract:**

Background: Pancreatic ductal adenocarcinoma (PDAC) remains one of the most lethal cancers. Given the currently limited therapeutic options, the definition of molecular subgroups with the development of tailored therapies remains the most promising strategy. Patients with high-level gene amplification of urokinase plasminogen activator receptor (*uPAR/PLAUR*) have an inferior prognosis. We analyzed the uPAR function in PDAC to understand this understudied PDAC subgroup’s biology better. Methods: A total of 67 PDAC samples with clinical follow-up and TCGA gene expression data from 316 patients were used for prognostic correlations. Gene silencing by CRISPR/Cas9, as well as transfection of *uPAR* and mutated *KRAS*, were used in PDAC cell lines (AsPC-1, PANC-1, BxPC3) treated with gemcitabine to study the impact of these two molecules on cellular function and chemoresponse. HNF1A and KRT81 were surrogate markers for the exocrine-like and quasi-mesenchymal subgroup of PDAC, respectively. Results: High levels of uPAR were correlated with significantly shorter survival in PDAC, especially in the subgroup of HNF1A-positive exocrine-like tumors. uPAR knockout by CRISPR/Cas9 resulted in activation of FAK, CDC42, and p38, upregulation of epithelial makers, decreased cell growth and motility, and resistance against gemcitabine that could be reversed by re-expression of uPAR. Silencing of *KRAS* in AsPC1 using siRNAs reduced uPAR levels significantly, and transfection of mutated *KRAS* in BxPC-3 cells rendered the cell more mesenchymal and increased sensitivity towards gemcitabine. Conclusions: Activation of uPAR is a potent negative prognostic factor in PDAC. uPAR and KRAS cooperate in switching the tumor from a dormant epithelial to an active mesenchymal state, which likely explains the poor prognosis of PDAC with high uPAR. At the same time, the active mesenchymal state is more vulnerable to gemcitabine. Strategies targeting either KRAS or uPAR should consider this potential tumor-escape mechanism.

## 1. Introduction

Pancreatic ductal adenocarcinomas (PDACs) are among the human tumors with the worst prognosis. Most PDAC patients are already at an advanced stage at diagnosis, and resection as the most effective treatment is only feasible in 20% of patients [1]. With gemcitabine as a baseline combined with FOLFIRINOX, next to albumin-bound paclitaxel, therapeutic options are limited [2,3,4]. The current clinical staging of PDAC cannot fully predict tumor behavior, and the prognosis of patients receiving the same treatment varies considerably. Therefore, it is essential to develop robust molecular classifications of PDAC for more tailored therapeutic approaches [5]. An increasing number of molecular and histological subtypes already define subtype-specific therapeutic vulnerabilities and provide the opportunity to supplement current pathological classifications. Recent studies discovered many PDAC subtype-specific markers connected to different clinical behavior; however, the three main subtypes remain classical, quasi-mesenchymal (QM-PDA), and exocrine-like [6].

Nevertheless, there is now good evidence that cancer cells preserve cellular plasticity [7,8]. Increased levels of urokinase-type plasminogen activator receptor (uPAR) are associated with early invasion, metastasis, and poor prognosis in many solid and hematological tumors, including PDAC [9,10,11]. uPAR is a GPI-anchored cell membrane receptor without an intracellular domain that mediates the degradation of extracellular matrix (ECM) components [12], including fibronectin and vitronectin [13]. It locally increases plasmin activity that facilitates cell migration. Interaction of uPAR with integrins occurs indirectly through stabilized binding to vitronectin [14]. This leads to intracellular activation of the Ras pathway, the focal adhesion kinase (FAK), and the Rho family small GTPase Rac (reviewed in [15]). PDAC is also one of the tumors with the highest frequency of *KRAS* mutations. KRAS has not only been shown to activate cell proliferation through RAF/MEK/ERK [16]; it has also been reported to regulate uPAR expression by AP1-dependent transactivation of the *uPAR* promoter [17]. Downregulation or blocking of uPAR causes activation of FAK, Src, CDC42, and p38, resulting in cell-cycle arrest and dormancy [18,19].

We have previously shown that 50% of PDACs show overexpression of uPAR due to low or high-level amplifications of the uPAR gene *PLAUR.* These tumors are associated with an inferior prognosis [20]. In this study, we functionally studied the role of uPAR in cell lines and validated the results in a cohort of 67 PDAC patients with clinical follow-up supplemented by TCGA data of 316 PDAC patient samples.

## 2. Materials and Methods

### 2.1. Human Tissue Samples

Tumor samples from 67 PDAC patients organized on a multi-tissue array (TMA) were used for immunohistochemical staining (clinical data are summarized in Table 1). The patient sample collection was approved by the ethics committee of the University Medical Center Göttingen (GÖ 912/15).

### 2.2. Immunohistochemistry

Immunohistochemical staining (IHC) of 2 µm paraffin sections was performed according to standard methods. Briefly, after deparaffinization in serially diluted alcohol and blocking endogenous peroxide in 0.3% hydrogen peroxide in PBS, antigen retrieval was performed at 95 °C in either a low or high-pH Envision FLEX target retrieval solution (Agilent, Santa Clara, CA, USA) using PT Link (Agilent). Subsequently, the stainings were incubated for 1 h with primary antibodies, followed by washing in PBS and incubation with the appropriate detection system for 30 min (Envision, Agilent). Antibodies were used at predetermined optimal dilutions (Appendix A) with the proper positive and negative controls. Staining was visualized by 3,3-diaminobenzidine tetrahydrochloride solution, counterstained with hematoxylin, dehydrated, and mounted in Pertex. Using an H-score, all tissue samples were evaluated for nuclear staining of p-p38, uPA, uPAR, and PAI1. The H-score was calculated by 3 × the percentage of the strongest staining signal + 2 × the percentage of a moderate signal + the percentage of a weak signal, resulting in a value range from 0 to 300. HNF1A and KRT81 were graded for “low” or “high” expression according to signal intensity. The optimal levels for the discrimination between high and low signals of uPAR, HNF1A, and KRT81 were determined using the cutoff finder [21].

### 2.3. Cell Culture and Transient Expression of uPAR and KRAS^G12C^

The human pancreatic cancer cell lines BxPC-3, AsPC-1, CAPAN-2, MIA PaCa-2, PATU8988T, and PANC-1 were obtained from the American Type Culture Collection (ATCC) (Appendix A). All cells were grown in RPMI-1640 medium (Gibco, Waltham, MA, USA), supplemented with 10% FCS (Gibco), 1% L-glutamine (Gibco), and 1% Penicillin/Streptomycin (Gibco) under humidified conditions at 37 °C and 5% CO_2_. PANC-1 was transfected with the pCMV-AC-GFP vector PLAUR (NM_002659) human-tagged ORF clone (Origene, Rockville, MD, USA), and BxPC-3 with the pCMV6-Entry-KRAS^G12C^ vector (Origene Technologies Inc., Rockville, MD, USA) using the X-tremeGENE HP DNA transfection reagent (Merck, Darmstadt, Germany). Transfected cells were selected with G418 (400 ng/µL). uPAR protein levels were tested by ELISA as described above. KRAS^G12C^-expressing cells were selected using 2µg/mL puromycin.

### 2.4. Generation of ASPC-1 uPAR Knockouts by CRISPR/Cas9

The uPAR CRISPR/Cas9 knockout strategy is shown in Appendix A. Cells were transfected with two CRISPR/Cas9 constructs, pCMV-Cas9-RFP (target site: 5′-GGACCCTGAGCTATCGGACTGG-3′), and pCMV-Cas9-GFP (target site: 5′-AGGTAACGGCTTCGGGAATAGG-3′) (Sigma-Aldrich, Darmstadt, Germany) using the X-tremeGENE HP DNA transfection reagent (Merck, Rahway, NJ, USA) according to the manufacturer’s instructions. After transient CRISPR/Cas9 activation, fluorescence-activated cell sorting (FACS) of GFP/RFP double-positive cells was performed for clone selection. PCR-screened clones for the gRNA target site or a potential deletion, as described later (Appendix A). Clones that were heterozygous for the deletion were further screened for specific gRNA target site mutations by Sanger sequencing (Appendix A).

### 2.5. Genomic PCR and Sanger Sequencing

The gRNA target sites were amplified with the primers GFP F: 5′-CTGTCCCCATGGAGTCTCAC-3′, GFP R:5′-CATCCAGGCACTGTTCTTCA-3′, RFP F: 5′-CTGGAGCTGGTGGAGAAAAG-3′, and RFP R: 5′-GGATTGGGATGATGATGAGG-3′ using MyTaq™ Mix (Bioline, London, UK) and the PCR products were analyzed via QIAxcel (Qiagen, Hilden, Germany). The PCR product was purified with ExoSAP-ITTM (Applied Biosystems, Foster City, CA, USA), and sequenced according to Sanger sequencing using the BigDye^®^ terminator v3.1 cycle sequencing kit (Applied Biosystems, Waltham, MA, USA). Sequences were analyzed using an ABI 3500 genetic analyzer (Applied Biosystems).

### 2.6. Cell Viability Assay

The CellTiter 96^®^ AQueous one-solution cell-proliferation assay (MTS, Promega, Madison, WI, USA) was performed according to the manufacturer’s recommendations. In brief, 1 × 10^4^ cells were grown in a 96-well format in 100 µL/medium and treated with indicated conditions over different periods, as described under results. Then, 20 µL of the MTS reagent was added and incubated for 1–3 h at 37 °C, and the absorbance was measured at 490 nm and 655 nm. Relative cell viability after treatment was calculated by normalizing each value by the mean of the untreated control replicates. Unless stated otherwise, all experiments were conducted by pretreating cells with 80 nM of the specific siRNA or inhibitors for 24 h and subsequent treatment with 0.1 µM gemcitabine for 72 h.

### 2.7. Wound Healing Assay

siRNA or mock-transfected cells were grown to almost 100% confluency before synchronizing the cells by decreasing FCS to 1% for 24 h. Wounds were created by scratching the cell monolayer with a 100 μL sterile pipette tip. Wound healing was monitored at 0, 24, and 48 h. Relative wound healing was calculated by measuring the mean distance at three defined positions of the scratch expressed as a percentage of the 0 h control.

### 2.8. siRNA Knockdown Experiments

siRNA transfection was performed using HiPerFect transfection reagent (Qiagen) as described elsewhere [22]. In brief, 80 nM of gene-specific or negative control siRNA (all Star Negative Control, Qiagen) was incubated with 12 µL HiPerFect in 100 µL transfection medium containing serum-free RPMI at RT for 20 min and added to freshly seeded cells (3 × 10^5^ cells). After 24 h or 48 h incubation, cells were further processed as indicated. siRNAs used were purchased from Qiagen and are summarized in Appendix A.

### 2.9. Protein Extracts, Western Blot Analyses, and uPAR Quantification by ELISA

Cells at 60–70% confluency were treated as indicated in the results section. Cells were washed in PBS and scraped in a 100 µL RIPA lysis buffer containing protease inhibitor cOmplete (Roche, Mannheim, Germany), PMSF (1 mM), and orthovanadate (1 mM). Total protein was quantified using a DC™ protein assay (Bio-Rad, Hercules, CA, USA). A total of 15 µg of proteins was separated using gradient SDS gels (4–20%, Bio-Rad) and blotted on nitrocellulose membranes by a Turbo Blot (Bio-Rad). Gene signals were detected as described before [23].

uPAR protein levels were determined by ELISA (DUP00, R&D Systems, Minneapolis, USA) according to the manufacturer’s protocol. In brief, cell lysates from 10^5^ to 10^6^ cells were 10-fold diluted in a RIPA lysis buffer, and 50 μL of cell lysates or standard was added to 100 μL of assay diluent RD1W solution. The samples were incubated for two hours at RT and washed four times with a 400 μL wash buffer. A total of 200 μL of human uPAR conjugate was added and incubated for 2 h at RT. After four washing steps, 200 μL of substrate solution was added and incubated for 30 min at RT protected from light before adding 50 μL of stop solution. The optical density was measured at 450 nm with a reference of 540 nm on a Tecan reader Infinite 200 Pro. uPAR concentrations were calculated for 10^6^ cells.

### 2.10. KRAS Activity Measurement

KRAS activity was quantified using the STA-400-K-T assay (Cell Biolabs) following the manufacturer’s instructions. In brief, 1 mg protein was subjected to raf1 RBD agarose beads and incubated at 4 °C for one h. Beads were pelleted, washed, and resuspended in 4× Laemmle buffer. KRAS activity was quantified by Western blotting of 20 µg supernatant protein.

### 2.11. Statistical Analysis

Statistical analysis and AUC estimation were performed using GraphPad 8.3.0. Data are shown as mean ± SEM. Two group comparisons were performed using Student’s *t*-test. Two-way ANOVA was applied to compare cell growth and resistance analyses. Survival was analyzed using the Kaplan–Meier test and significance was evaluated using the log-rank (Cox–Mantel) test. A *p*-value of < 0.05 was considered significant (* = *p* < 0.05, ** = *p* < 0.01, *** = *p* < 0.001).

## 3. Results

### 3.1. uPAR Protein and mRNA Expression Levels Have Prognostic Significance in PDAC

Our previous study showed that *uPAR* gene amplification in PDAC correlates with poor prognosis [24]. Immunohistochemical (IHC) staining for uPAR, its ligand uPA, and the inhibitor PAI1 in a clinical cohort of 67 patients (Figure 1, Table 1) also confirmed a prognostic relevance of uPAR on the protein level. Patients with high uPAR expression (*n* = 46) had significantly shorter overall survival (OS) than patients with low uPAR levels (*n* = 23) (median survival 320 days in uPAR^high^ vs. 603 days in uPAR^low^ patients, log-rank (Cox–Mantel) test, *p* = 0.0273) (Figure 1b) [25]. Using gene expression data from two TCGA datasets including 312 PDAC patients [26,27], patients with high uPAR mRNA expression had a significantly reduced OS compared to patients with tumors of low expression (log-rank (Cox–Mantel) test, *p* = 0.0099) (Figure 1c). IHC did not reveal any significant difference in OS for uPA and PAI1 (Appendix A); however, on the transcriptional level, high expression of both uPA and PAI1 showed a significantly decreased OS (Appendix A).

### 3.2. Generation of CRISPR/Cas9 uPAR Knockout Clones in AsPC-1 Cells

Next, we wanted to investigate the molecular function of uPAR in PDAC cells. Therefore, we measured the uPAR protein expression levels by ELISA in six PDAC cell lines with known gene mutation status of *KRAS*, *TP53*, and *PIK3CA* as described in the Material and Methods section (Appendix A and Appendix A). We then generated *uPAR* knockout clones of the cell line with the highest uPAR expression (AsPC-1), using two gRNAs directed against *uPAR* exons 3 and 4 (Appendix A). Two clones with homozygous functional *uPAR* knockout (KO#1 and KO#2), carrying a deletion on one allele and a gRNA target-site-specific frameshift mutation on the other, revealed a virtually absent uPAR protein (Appendix A).

### 3.3. uPAR Influences Cell Growth, Cellular Plasticity, and the Response to Gemcitabine in AsPC-1 (KRAS^G12D^)

Functional roles of uPAR have been described in cell proliferation, migration, and cellular plasticity [28,29,30,31]. Both AsPC-1 *uPAR^−/−^* clones showed a significant decrease in growth and migration capacity compared to the AsPC-1 WT controls (Figure 2a,b). To evaluate the role of uPAR in cellular plasticity, we immunoblotted nine markers involved in epithelial–mesenchymal transition (EMT) (Figure 2c).

Western blot revealed a marked upregulation of epithelial markers E-cadherin and β-catenin. While the transcription factor Slug was slightly upregulated, Snail and TCF8/ZEB1, together with claudin and ZO1, showed a decreased expression, further indicating the mesenchymal to epithelial transition (MET) in *uPAR^−/−^* clones compared to AsPC-1 WT (Figure 2c). In accordance with this phenotype, we detected a marked increase in chemoresistance against up to 1 µM gemcitabine in *uPAR^−/−^* cells (Figure 2d).

### 3.4. Depletion of uPAR Activates FAK, CDC42, and p38 and Induces Autophagy

uPAR signaling has been described to involve FAK, Src, CDC42, p38, autophagy, and RAS signaling [32]. In addition, Wu et al. reported that FAK signaling contributes to intrinsic gemcitabine chemoresistance in pancreatic cancer cell lines [33]. By immunoblotting, we detected the activation of FAK, CDC42, p38, and LC3B, while ERK was inactivated in AsPC-1 *uPAR^−/−^* cells (Figure 3a). The influence of FAK on Ras signaling has been described before [34]. However, in cells with aberrant KRAS activation, FAK-Ras regulation seems to be disturbed.

Knockdown of FAK in *uPAR^−/−^* cells using siRNAs led to decreased phosphorylation of CDC42, p38, and LC3B, and reactivation of ERK (Figure 3b). The diminished FAK activity also partially restored the sensitivity towards gemcitabine (Figure 3c and Appendix A). Knockdown of CDC42 and p38 also reactivated ERK, decreased LC3B, and increased gemcitabine sensitivity (Figure 3d–g and Appendix A). This indicates that CDC42 and p38 suppress ERK activity downstream of KRAS in the absence of uPAR.

### 3.5. Re-expression of uPAR Restores the Migratory Capability and Gemcitabine Sensitivity of uPAR^−/−^ Cells

To evaluate whether uPAR re-expression could restore the WT phenotype, *uPAR^−/−^* cells were transfected with a human *uPAR* gene expression vector as described in the Material and Methods section. This recovered uPAR protein levels (Appendix A) and significantly enhanced migratory capacity (Figure 3h). uPAR re-expression also recovered gemcitabine sensitivity and induced resistance against the p38 inhibitor JX401 (Figure 3i and Appendix A). Pharmacological inhibition of ERK with SCH772948 reduced gemcitabine sensitivity only in uPAR WT but not in AsPC-1 *uPAR^−/−^* cells (Figure 3j and Appendix A). Together, this indicates that uPAR mediates gemcitabine sensitivity in an ERK-dependent manner.

### 3.6. Resistance against Gemcitabine in AsPC-1 uPAR^−/−^ Cells through Autophagy

The autophagy marker LC3B was induced in AsPC-1 *uPAR^−/−^* cells. Autophagy promotes tumor cell survival and contributes to chemoresistance [35]. Increased autophagy has been described to be responsible for the resistance of PDAC to gemcitabine that could be partially reversed by specific inhibitors [36]. To investigate whether increased autophagy in *uPAR^−/−^* clones was responsible for the observed gemcitabine resistance, we inhibited autophagy with 3-methyladenine (3-MA) or chloroquine (CQ). Both inhibitors significantly restored sensitivity towards gemcitabine in AsPC-1 *uPAR^−/−^* but not in AsPC-1 *WT* (Figure 3k and Appendix A).

### 3.7. uPAR and Mutated KRAS Cooperate in Maintaining a Mesenchymal Phenotype

To evaluate the interplay of uPAR and mutated *KRAS* in response to gemcitabine, we used the *KRAS* WT cell line BxPC-3 (uPAR high), the *KRAS* mutant cell line AsPC1 (uPAR high), and the *KRAS* mutant cell line PANC-1 (uPAR low) (Figure 4a and Appendix A). AsPC1 responded best towards gemcitabine, PANC-1 showed a medium response, and BxPC3 was the most resistant cell line (Figure 4b). KRAS has been described to induce uPAR expression [17]. Silencing of *KRAS* in AsPC1 using siRNAs reduced uPAR levels significantly (Figure 4c). Silencing of *KRAS* in AsPC1 reduced the response towards gemcitabine whereas the expression of mutated *KRAS* in BxPC-3 cells increased gemcitabine sensitivity. Transfection of uPAR in PANC-1 likewise increased gemcitabine sensitivity (Figure 4d). uPAR and mutated KRAS switched cells to a mesenchymal phenotype **(**Figure 4e), at the same time promoting activation of MEK and ERK and suppressing FAK and CDC42 signaling (Figure 4f).

### 3.8. uPAR Modulates the Clinical Risk in Different PDAC Subgroups

Noll et al. [8] published HNF1A as a surrogate marker for the exocrine-like PDAC subtype and expression of keratin 81 (KRT81) as a marker for the quasi-mesenchymal (QM) type. Tumors negative for both markers (DN) were enriched for the classical PDAC subtype. We wanted to know if tumors with high uPAR expression segregate with one of these subtypes. In our own cohort of 57 patients with clinical follow-up, *n* = 31 (54%) showed expression of HNF1A, *n* = 19 (33%) were positive for KRT81, and *n* = 7 (12%) were DN. Because the DN group was too small, we excluded it from further analysis. The exocrine-like group consisted of 21 uPAR low and 10 uPAR high cases, and the QM group contained 9 uPAR low and 10 uPAR high cases. Survival analysis was supplemented by gene expression data from the two TCGA cohorts (*n* = 82 cases HNF1A high vs. *n* = 85 cases KRT81 high).

The overall survival of patients with HNF1A-positive exocrine-like PDAC was significantly longer than patients with KRT81-positive QM tumors (*p* < 0.0001, Figure 4g). In the HNF1A-positive cohort, tumors with low levels of uPAR had a significantly better outcome than tumors with high expression and the mortality curve even reached a plateau after 1000 days, indicating long-term survival of some patients. In the KRT81^high^ QM and DN group, there was a trend towards longer survival in patients with tumors with low levels of uPAR that did not reach statistical significance (Figure 4h,i and Appendix A), indicating that the prognostic impact of uPAR may vary among different molecular subgroups.

## 4. Discussion

PDAC remains one of the human tumors with the highest mortality. uPAR is associated with early invasion, metastasis, and poor prognosis in many solid and hematological tumors [9,10,11]. We have previously shown that PDAC with high-level gene amplifications of *uPAR* have a particularly poor prognosis [24]. We here show in our cohort of 67 samples and in 168 PDAC samples from the TCGA database that overexpression of uPAR on the mRNA and protein level is also associated with significantly shorter OS. Importantly, although our data suggest that high expression of uPAR is an adverse prognostic factor in all PDAC, its negative impact on survival is more pronounced in some molecular subgroups (especially in exocrine-like tumors) than in others.

uPAR has been described to act through its vitronectin-mediated interaction with integrins to transmit mechanical forces across the cell membrane [37,38,39]. The ECM–integrin interaction mediates the intrinsic chemoresistance of cancer cells [40], a phenomenon that has also been called cell-adhesion-mediated drug resistance (CMDR). CMDR has been explained by the strong binding of integrins to the ECM, which activates FAK. Integrin and EGFR signaling activates FAK and influences adhesion, motility, and cell growth [41,42]. FAK has seemingly paradoxical roles in cell migration and metastasis [43]. FAK is a ubiquitously expressed tyrosine kinase that localizes at focal adhesion complexes and transmits adhesion- and growth-factor-dependent signals into the cell [34,43,44]. In contrast to normal cells where FAK is a positive regulator of cell migration and proliferation [45], tumors with constitutive growth factor signaling (such as EGFR) or *RAS* mutations and consecutive high intrinsic levels of ERK utilize FAK as a negative regulator of cell migration through ERK-dependent dephosphorylation of particular FAK tyrosine residues [43,46]. Constitutive activation of FAK has also been proposed to contribute to the intrinsic chemoresistance against gemcitabine in the pancreatic cancer cell line AsPC-1 [9,33]. We here show that uPAR knockout in AsPC1 cells leads to induction of FAK, Src, CDC42, and p38, as well as chemoresistance towards gemcitabine. Our data further show that this chemoresistance is mediated through p38-induced autophagy. Numerous early clinical trials [47] have shown significant antitumor activity with tolerable toxicity of the autophagy inhibitor chloroquine, in combination with other cytotoxic chemotherapies in a variety of solid cancers, including colorectal and renal cell carcinomas [48]. A randomized clinical phase II trial in 102 PDAC patients treated with gemcitabine and nab-paclitaxel with or without CQ showed no difference in progression-free survival. Still, the authors proposed that preoperative CQ might increase curative resection rates [49].

A total of 90–95% of PDACs harbor activating mutations of *KRAS* that are thought to occur early in carcinogenesis [16]. Mutated *KRAS* is a potent oncogenic driver that promotes cell proliferation and migration by activating the downstream MAP kinases ERK1/2 [50]. KRAS has not only been reported to induce uPAR expression by AP-1-dependent transactivation of the *uPAR* promoter [17], but also mediates FAK dephosphorylation [43]. We here show that a) constitutively active KRAS induces uPAR and b) KRAS and uPAR cooperate in promoting a mesenchymal cell phenotype by activating MEK/ERK signaling and by the suppression of FAK/CDC42/p38 signaling. At the cellular level, this mesenchymal state implies increased cell proliferation and migration as a possible explanation of the poor prognosis of tumors with high levels of uPAR. At the same time, it also implies suppressed cellular dormancy via FAK signaling and p38-mediated autophagy, thus rendering the cells more vulnerable to gemcitabine. These observations highlight a potential therapeutic dilemma that applies both to KRAS and uPAR as emerging targets. Although recent studies propose uPAR as a good candidate for antibody-targeted therapy in cancer [51,52,53,54,55], our results show that these treatments could, at the same time, induce cellular dormancy and render the tumor more resistant to chemotherapy (such as gemcitabine). Tailored strategies should consider this resistance by adding autophagy inhibitors, such as chloroquine, to the regimens.

## 5. Conclusions

In summary, we have confirmed uPAR as a potent modulating prognostic factor, especially in the large molecular subgroup of exocrine-like tumors. uPAR cooperates with mutated KRAS in the important switch between an active mesenchymal vs. a dormant epithelial cellular phenotype. By keeping tumor cells in the active mesenchymal state, uPAR promotes *KRAS*-driven proliferation and cell migration as a likely explanation for the poor prognosis of PDAC with high expression of uPAR. At the same time, this active mesenchymal state renders tumor cells more vulnerable to chemotherapy such as gemcitabine. Targeting either uPAR or KRAS could induce cellular dormancy and autophagy, thus leading to relative chemoresistance and limited therapeutic efficacy. Emerging clinical trials should take this possibility into account.

## Figures and Tables

**Figure 1 cancers-15-01587-f001:**
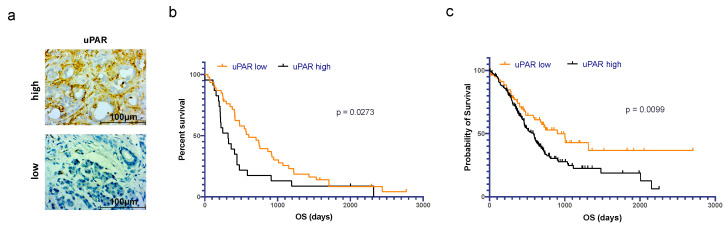
Prognostic significance of uPAR expression in PDAC patients. (**a**) Exemplary immunohistochemical staining of PDAC with high vs. low expression of uPAR. (**b**) Statistically significant difference in OS for *n* = 67 PDAC patients with high (orange, *n* = 45) vs. low (black, *n* = 22) immunohistochemical expression of uPAR. (**c**) Statistically significant difference in OS for *n* = 83 PDAC patients with high (black) vs. *n* = 219 patients with low (orange) expression levels of uPAR mRNA (source: TCGA dataset).

**Figure 2 cancers-15-01587-f002:**
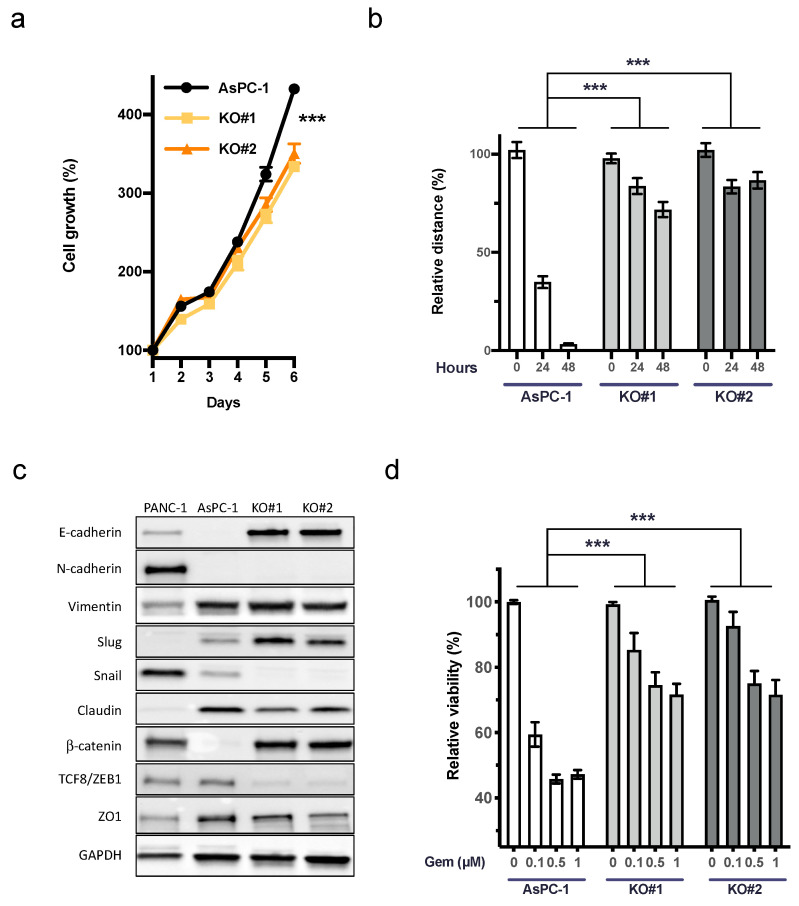
Decreased cell growth, motility, and response to gemcitabine of AsPC uPAR knockout clones. (**a**) Cell growth analysis (6 days) of *uPAR^−/−^* clones with a significantly slower proliferation rate than WT controls (*n* = 3). (**b**) Reduced migratory capacity of AsPC-1 *uPAR^−/−^* clones compared to *uPAR^WT^* cells (*n* = 3). (**c**) Western blot analysis of 9 epithelial and mesenchymal markers in PANC-1, AsPC-1, and *uPAR^−/−^* clones indicated *mesenchymal to epithelial transition* (MET) in *uPAR^−/−^* cells. Uncropped Western blot images available in Appendix A (**d**) Increased resistance of *uPAR^−/−^* clones to gemcitabine treatment (0.1, 0.5 and 1 µM) for 72 h (*n* = 4 biological replicates). (KO#1 and KO#2, *uPAR^−/−^* clones) (*** *p* < 0.001).

**Figure 3 cancers-15-01587-f003:**
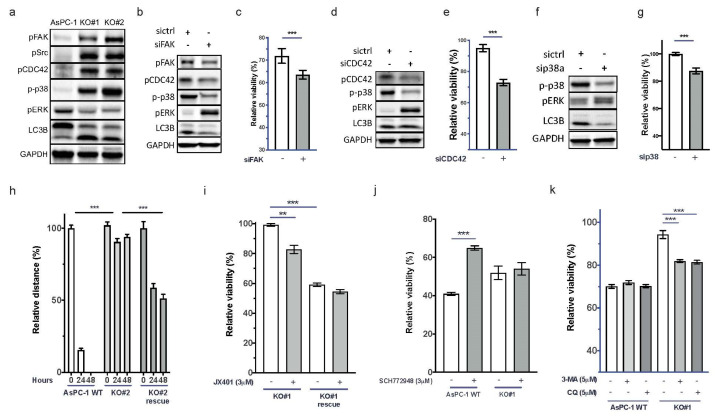
uPAR regulates CDC42, p38, LC3B, and ERK activity. (**a**) Immunoblot showing increased signals for pCDC42, pSrc, p-p38, pERK, and LC3B in KO#1 and KO#2.). Uncropped Western blot images could be found at in Appendix A. Restoration of (**b**) the wild-type signaling phenotype after FAK siRNA knockdown in AsPC-1 *uPAR^−/−^* cells and (**c**) of the response to gemcitabine (*n* = 4). Uncropped Western blot images could be found at in Appendix A (**d**) Knockdown of CDC42 by siRNA and (**e**) response to gemcitabine and (**f**) siRNA knockdown of p38 and (**g**) the corresponding gemcitabine response. Uncropped Western blot images available in Appendix A. (**h**) Increased cellular motility after transient uPAR expression in KO#2 (KO#2 rescue) compared to AsPC-1 *uPAR^−/−^* cells (*n* = 4). (**i**) Gemcitabine (0.1 µM) and combinational treatment with the p38 inhibitor JX401 for 72 h in AsPC-1 *uPAR^−/−^* and *KO#1* uPAR rescue cells (*n* = 3). (**j**) Gemcitabine (0.1 µM) and combinational treatment with the ERK inhibitor SCH772948 in uPAR WT and AsPC-1 *uPAR^−/−^* cells (KO#1). (**k**) Treatment of AsPC-1 WT and AsPC-1 *uPAR^−/−^* (KO#1) with either gemcitabine (0.1 µM) or in combination with the autophagy inhibitors 3-MA (5 µM) or CQ (5 µM). Relative viability is shown in response to gemcitabine and in combination with siRNA/inhibitors. (*n* = 4 biological replicates (** *p* < 0.01, *** *p* < 0.001).

**Figure 4 cancers-15-01587-f004:**
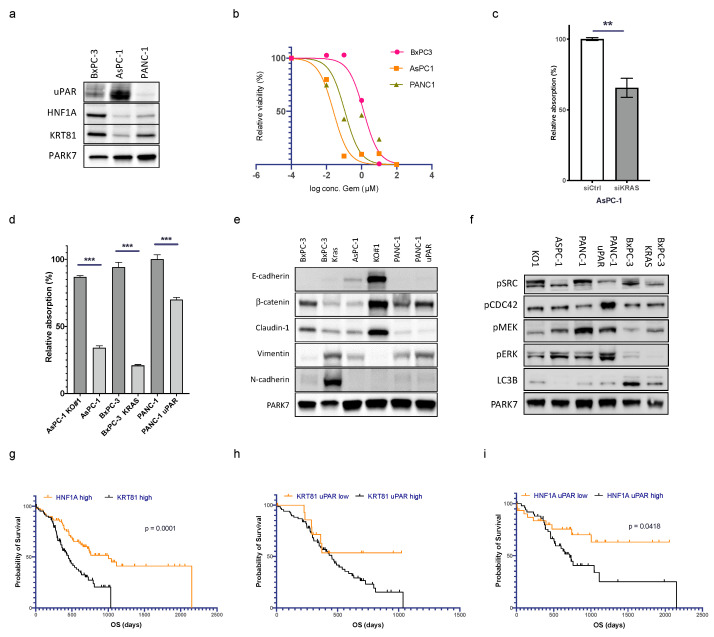
uPAR and mutated KRAS cooperate in maintaining a mesenchymal phenotype that also regulates gemcitabine sensitivity. (**a**) Immunoblot showing uPAR, HNF1A, and KRT81 expression in BxPC-3, AsPC-1, and PANC-1. Uncropped Western blot images available in Appendix A (**b**) IC50 of gemcitabine treatment (0–100 µM, 72 h) in BxPC-3 (1.323 µM), AsPC-1 (0.025 µM), and PANC-1 (0.112 µM). (**c**) uPAR levels after KRAS siRNA knockdown in AsPC-1 (*n* = 3 biological replicates, ** Student’s *t*-test, *p* < 0.01, *** Student’s *t*-test, *p* < 0.001). (**d**) Gemcitabine response (0.125 µM, 72 h) in AsPC-1 WT, AsPC-1 *uPAR^−/−^* (KO#1), BxPC-3 (*KRAS* WT), BxPC-3 (*KRAS*^mut^), PANC-1 (uPAR^low^), and PANC-1 (uPAR^high^). (**e**) Immunoblot of protein lysates from the same cell lines for EMT markers and (**f**) pFAK, pCDC42, p-p38, pMEK, p-ERK, and LC3B. Kaplan–Meier curves using mRNA expression data of PDAC from the TCGA cohort. Uncropped Western blot images available in Appendix A. (**g**) Comparison of PDAC with high expression of *HNF1A* vs. *KRT81*. (**h**) *KRT81^high^* tumors with high vs. low expression of *uPAR* and (**i**) HNF1A*^high^* tumors with high vs. low expression of *uPAR* (log-rank test, *p* < 0.05).

**Table 1 cancers-15-01587-t001:** Clinical data summary.

Patients	67
Male (%)	37 (55)
Female (%)	30 (45)
Age median (range)	68 (44–84)
Tumor grade (G)	
1–2 (%)	6 (9)
2–3 (%)	41 (61.1)
3–4 (%)	20 (29.9)
Tumor stage (TNM)	
T 1 (%)	1 (1.5)
T 2 (%)	3 (4.5)
T 3 (%)	58 (86.6)
T 4 (%)	5 (7.4)
N 0 (%)	14 (20.9)
N 1–3 (%)	53 (79.1)
Median follow-up time (range) [day]	417 (4–2768)
Reported deaths (%)	62 (92.5)

## Data Availability

The data presented in this study are available in the article, the Appendix A and at TCGA (https://www.cbioportal.org, accessed on 30 January 2023: Pancreatic Adenocarcinoma, Firehose Legacy and PanCancer Atlas).

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
