# Peer review of "Urokinase-Type Plasminogen Activator Receptor (uPAR) Cooperates with Mutated KRAS in Regulating Cellular Plasticity and Gemcitabine Response in Pancreatic Adenocarcinomas"

_cancers, 2023, doi:10.3390/cancers15051587_

Round 1

Reviewer 1 Report

This is a very thorough follow up investigation to your prior publication analyzing the role of uPAR in PDAC. 

The background section covered the current limitations in therapy of PDAC and your prior work instigating this follow up analysis. 

Methods were clear and followed standard practice. no novel procedures were employed. 

Results were clear and images/figures summarized your findings well. 

Discussion succinctly summarized your findings and the significant limitations in pursuing this as a target. 

The one notable limitation of this study is that it is evaluating chemosensitivity to gemcitabine alone. It has been many years since gemcitabine has been the standard of care. Although one could extrapolate that gemcitabine sensitivity represents all chemotherapy, I think limiting your analysis to gem alone limits next steps clinically.

Author Response

We thank Reviewer #1 for his feedback. We are aware of the limits analyzing gemcitabine only.
Although it was not possible to show this in the frame of this study, it is very likely that the uPAR-KRAS axis will be also relevant for other treatments, since the main effects (switch from mesenchymal to epithelial state, induction of autophagy etc.) occur independently of specific treatment.

Reviewer 2 Report

Dear authors,

Please see below minor changes required:

1. The text has many errors when it comes to Figure labeling. For example:

-Check legend of Supplementary Figure 1. Is everything labelled correctly?

-Line 231. Do you mean Supplementary Figure 2F-H?

-Check x axis labeling of figure 3K

2. Can the authors please explain why they chose to do a transient instead of a stable expression and knockdown? Were the proteins sufficiently silenced/expressed during the duration of functional assays?

3. Minor grammatical/styling errors were spotted e.g line 117 (capital letter), Lines 222 and 233, 293, 316 etc should be bold,

4. I found the presentation of the migration data in Fig.2B confusing. The authors need to explain a bit better on how this graph was created. The knockout cells traveled less than the controls correct? Also images from the scratch assay even in supplementary would be useful.

5. In Supplementary Figure 2H is there statistical significance between knockout and rescue?

6. In reference to panel 3J they mention that "Pharmacological inhibition of ERK with SCH772948 reduced gemcitabine sensitivity only in uPAR WT but not in AsPC-1 uPAR-/- cells". Should the comparison then be between the second and 4th bar as shown on the graph?

7. Can the authors comment on the fact that in Fig.4A the PANC1 cell line shows low expression of uPAR but the sensitivity to Gem is intermediate compared to the other two cell lines?

Author Response

We thank Reviewer 2 his feedback and for several significant remarks. Below you find the point-by-point responses and explanations.

1. The text has many errors when it comes to Figure labeling.
We apologize for the incorrect Figure labeling and changed as following:

-Check legend of Supplementary Figure 1. Is everything labelled correctly?
Legend of Fig. S1 was checked and re-phrased for clarity.

-Line 231. Do you mean Supplementary Figure 2F-H?
“Supplementary Figure 2 f-h” as given in the figure was corrected.

-Check x axis labeling of figure 3K
We checked the labeling of the x-axis of Figure 3k but did not find any mistakes. To make the figure better understandable we adjusted the labels and added an additional remark in the figure legend.

2. Can the authors please explain why they chose to do a transient instead of a stable expression and knockdown? Were the proteins sufficiently silenced/expressed during the duration of functional assays?

We agree that stable knockdown and overexpression are certainly preferable. We used siRNAs for some of our experiments due to their less demanding experimental handling, since they are still well-accepted in the science field. To strengthen the specific effect, we always used two different siRNAs targeting the same gene and compared them to an arbitrary siRNA negative control to exclude off-target effects. We controlled the protein expression of the targeted gene 24 hours after transfection for a significant knockdown by western blotting in parallel to drug treatment. For functional assays, the silencing was controlled at 48 hours. We also would like to point out that even if there is a possible rebound effect when siRNAs are diluted out during the assay, we would still have a pulse-chase effect of the knockdown at 24 and 48 hours.

For the overexpression of uPAR and aberrant KRAS we have chosen puromycin and hygromycin selectable expression vectors. These do not lead to a stable insertion of the gene in the chromosomal DNA, but gene-expressing cells can be selected by the antibiotics and ensure a homogeneous transgene-expressing cell population.

3. Minor grammatical/styling errors were spotted e.g line 117 (capital letter), Lines 222 and 233, 293, 316 etc should be bold,

We intensely re-evaluated grammar and styling and hope to meet now the recommendations.

4. I found the presentation of the migration data in Fig.2B confusing. The authors need to explain a bit better on how this graph was created. The knockout cells traveled less than the controls correct? Also images from the scratch assay even in supplementary would be useful.

We like to thank the reviewer for this important comment. It is correct that uPAR-knock out cells (in good agreement with their more epithelial morphology) migrated less than wild type cells We added an additional Supplementary Figure 2i in order to better illustrate this observation.

5. In Supplementary Figure 2H is there a statistical significance between knockout and rescue?

Due to high variations of the duplicate measurements, significance could not be accurately evaluated. However, the measurements indicated a 2 to 17-fold higher uPAR expression in AsPC-1 rescue in comparison to AsPC-1 WT.

6. In reference to panel 3J they mention that "Pharmacological inhibition of ERK with SCH772948 reduced gemcitabine sensitivity only in uPAR WT but not in AsPC-1 uPAR-/- cells". Should the comparison then be between the second and 4th bar as shown on the graph?

The difference was calculated in relation to the untreated cells (either uPAR WT or knockdown). A significant difference was only found for ASPC wildtype, but not for the knockdown clones. In our view, the figure is therefore correct.

7. Can the authors comment on the fact that in Fig.4A the PANC1 cell line shows low expression of uPAR but the sensitivity to Gem is intermediate compared to the other two cell lines?

As we describe in the manuscript, uPAR and mutated KRAS cooperate in regulating the mesenchymal vs. epithelial cellular phenotype that also determines gemcitabine response. In Fig. 4A, the cell line with the highest gemcitabine sensitivity was ASPC-1 (uPAR high and KRAS mut), followed by PANC1 (uPAR intermediate and KRAS mut), followed by BxPC3 (uPAR low and KRAS WT). We think that this observation further supports the key hypothesis behind this manuscript.